# Efficacy of Ursolic Acid-Enriched Water-Soluble and Not Cytotoxic Nanoparticles against Enterococci

**DOI:** 10.3390/pharmaceutics13111976

**Published:** 2021-11-21

**Authors:** Anna Maria Schito, Debora Caviglia, Gabriella Piatti, Alessia Zorzoli, Danilo Marimpietri, Guendalina Zuccari, Gian Carlo Schito, Silvana Alfei

**Affiliations:** 1Department of Surgical Sciences and Integrated Diagnostics (DISC), University of Genoa, Viale Benedetto XV, 6, 16132 Genoa, Italy; amschito@unige.it (A.M.S.); Cavigliad86@gmail.com (D.C.); gabriella.piatti@unige.it (G.P.); giancarlo.schito@unige.it (G.C.S.); 2Stem Cell Laboratory and Cell Therapy Center, IRCCS Istituto Giannina Gaslini, via Gerolamo Gaslini 5, 16147 Genoa, Italy; alessiazorzoli@gaslini.org (A.Z.); danilomarimpietri@gaslini.org (D.M.); 3Department of Pharmacy, University of Genoa, Viale Cembrano 4, 16148 Genoa, Italy; zuccari@difar.unige.it

**Keywords:** fourth-generation polyester-based lysine-modified dendrimer, physical encapsulation, ursolic acid (UA), Gram-positive MDR isolates, MICs, time-kill experiments, cytotoxicity on human keratinocytes

## Abstract

Ursolic acid (UA), a pentacyclic triterpenoid acid found in many medicinal plants and aromas, is known for its antibacterial effects against multi-drug-resistant (MDR) Gram-positive bacteria, which seriously threaten human health. Unfortunately, UA water-insolubility, low bioavailability, and systemic toxicity limit the possibilities of its application in vivo. Consequently, the beneficial activities of UA observed in vitro lose their potential clinical relevance unless water-soluble, not cytotoxic UA formulations are developed. With a nano-technologic approach, we have recently prepared water-soluble UA-loaded dendrimer nanoparticles (UA-G4K NPs) non-cytotoxic on HeLa cells, with promising physicochemical properties for their clinical applications. In this work, with the aim of developing a new antibacterial agent based on UA, UA-G4K has been tested on different strains of the *Enterococcus* genus, including marine isolates, toward which UA-G4K has shown minimum inhibitory concentrations (MICs) very low (0.5–4.3 µM), regardless of their resistance to antibiotics. Time-kill experiments, in addition to confirming the previously reported bactericidal activity of UA against *E. faecium*, also established it for UA-G4K. Furthermore, cytotoxicity experiments on human keratinocytes revealed that nanomanipulation of UA significantly reduced the cytotoxicity of UA, providing UA-G4K NPs with very high LD_50_ (96.4 µM) and selectivity indices, which were in the range 22.4–192.8, depending on the enterococcal strain tested. Due to its physicochemical and biological properties, UA-G4K could be seriously evaluated as a novel oral-administrable therapeutic option for tackling difficult-to-treat enterococcal infections.

## 1. Introduction

For many years, antibiotics have made it possible to counteract serious bacterial infections and to perform various medical and surgical procedures in greater safety, thanks to their protection. Unfortunately, over the past 20 years, these drugs have been used unscrupulously, leading to the selection of an increasing number of multidrug-resistant bacteria (MDR), which greatly threaten human health [1].

Today, MDR pathogens of both Gram-positive and Gram-negative species cause infections against which traditional antibiotics are no longer effective and cause an increasing number of deaths in hospitals, long-term care facilities, and community settings. The surveillance of antibiotic resistance is a global public health concern [2,3,4,5,6,7,8], and the search for new antibacterial substances, which act by mechanisms other than those of existing antibiotics, and with a low tendency to develop resistance, is at present one of the greatest challenges for researchers [9].

Especially Gram-negative pathogens, such as *Acinetobacter baumannii*, *Pseudomonas aeruginosa,* and *Stenotrophomonas maltophilia,* are now emerging as clinically relevant superbugs, contributing significantly, with their worrying levels of resistance, to numerous therapeutic failures [1]. However, antibiotic resistance has also become a major problem in treating infections caused by many Gram-positive bacteria, including penicillin-resistant *Streptococcus pneumoniae*, methicillin-resistant *Staphylococcus aureus* (MRSA) and *S. epidermidis* (MRSE), and enterococcal species, such as *Enterococcus faecium* and *E. faecalis*, which express high-level resistance to aminoglycosides and/or resistance to vancomycin (VRE) [10]. Often, these strains become tolerant to currently available agents, thus requiring innovative therapeutic strategies, including the use of alternative and nonconventional drugs, alone or in combination, and the development of new drugs. Numerous studies have established that various beneficial bioactivity, including antibiotics effects, can result from the daily intake of natural products with a normal diet or from the administration of isolated natural compounds found in plants [11]. Nowadays, natural antibacterial compounds represent a significant alternative for the pharmaceutical, food, and cosmetic industries, as they meet the needs of ‘green consumerism’ while possessing excellent antibacterial activity.

Aromatic plants are the main sources of natural antibacterial products, including monoterpenes, sesquiterpenes, caffeic acid, luteolin, rosmarinic acid, hispidulin, flavonoids, oleanolic acid (OA) and ursolic acid (UA) [11]. The latter can be obtained in high yield also by vegetable by-products, such as apple pomace by ultrasonic-assisted extraction [12]. OA and UA, which are isomeric pentacyclic triterpene acids, often coexist in medicinal plants. Several works have been published regarding the study of their solubility in different aqueous mixtures of solvents and at different temperatures to optimize the extraction procedures, their separations, and purification [13,14,15].

UA responds to the chemical name 3β-hydroxy-urs-12-en-28-oic-acid (PubChem CID:64945, CAS:77–52-1) (Figure 1). It is a pentacyclic terpenoid that has considerable therapeutic potential and has aroused great interest in recent years [2,16,17,18,19,20].

UA is considered a promising compound for cancer prevention and therapy, as it influences cell signalling pathways, inhibiting enzyme activity, inducing apoptosis, and reducing tumor growth. It is abundant in the plant surface extracts [21,22,23,24,25,26] and showed potent antibacterial activity against several Gram-positive bacterial species [2,16,25,27], such as *S. aureus, E. faecalis, S. mutans, S. sobrinus* and *Mycobacterium tuberculosis* [28,29,30].

UA can be used synergistically with antibiotics for enhancing their activity [31,32,33] and has proven to be an efficient agent to disperse the biofilm generated by *S. aureus* [34] and to inhibit the formation of biofilm by MRSA isolates [35]. Although the precise mechanisms underlying these findings are not known in detail, a study on the mode of action of UA against MRSA reported initial irreversible damage to bacterial membrane integrity, followed by inhibition of protein synthesis and the metabolic pathway [36]. Recently, by molecular docking, it has been hypothesized that UA can modulate the amount of ATP by inhibiting the hydrolysis of ATP (in MRSA) as polymyxins [37].

Unfortunately, UA also exhibited unwanted features. Reports revealed that UA can trigger unwanted phenomena under certain conditions and can be cytotoxic to human cells [38,39]. In addition, the insignificant solubility and low stability of UA in aqueous medium, as well as its very poor bioavailability in vivo, make it practically non-administrable and considerably hinder its therapeutic application [40,41,42,43]. Therefore, the development of new water-soluble UA formulations capable of overcoming these disadvantages is urgently needed. In this regard, the use of NPs in medicine is an ever-growing research field, as evidenced by the huge number of scientific publications and the considerable number of formulations based on NPs registered for clinical trials [44]. Nanoparticulate systems, including liposomes, micelles, polymer-based NPs, dendrimer NPs, etc., have attracted the interest of many researchers, and numerous systems have been developed [44]. The prepared NPs must possess several requirements, such as being non-toxic and biocompatible, having stealth properties and reduced immunogenicity, being capable of transporting a large number of drug moles, and delivering them to the target site in a sustainable way.

In this regard, considerable progress has been made in the design of the ideal drug carriers, as evidenced by the Food and Drug Administration (FDA) approval obtained by some nanocarriers [44]. Moreover, several properties, such as biocompatibility, drug loading capacity, and site-specificity of drug release, need to be further improved.

Among the nano-systems designed to address these objectives, mainly polymer-based NPs have attracted the interest of scientists due to the versatile characteristics of polymers that allow adjustment of the physicochemical and biological properties of the corresponding nanocarriers. Consequently, several species of polymers have been developed to solubilize and formulate multifunctional drug carriers with properties adapted to the considered application [45]. In this contest, several strategies, based on nanotechnology, are reported in the literature aimed at increasing the water solubility of UA, such as its conversion into nanocrystals, the production of solid dispersion forms or nanoparticles (NP), its encapsulation or chemical conjugation to dendrimers, its absorption in mesoporous silica-based nanosphere (MSN), or its co-dissolution with lipids [41]. However, many of the approaches developed to solubilize UA have involved the use of high amounts of organic solvents, co-solvents (PEG, glycerol), stabilizers, surfactants, or emulsifiers (polaxamers), which can become toxic to humans [46].

In this scenario, with the final aim of making the clinical administration of UA feasible, we have recently increased its poor solubility in water and cancelled its residual toxicity on HeLa cells, trapping it in a cationic fourth-generation, not cytotoxic polyester-based dendrimer (G4K) containing lysine [47]. Highly water-soluble UA-loaded NPs (UA-G4K NPs) were obtained (Figure 1), characterized by a high drug loading (DL%) and encapsulation efficiency (EE%), and capable of a UA sustained release profile governed by diffusion mechanisms [47], thus meeting the requirements above-mentioned, which NPs for biomedical application should possess.

Moreover, concerning the cytotoxicity experiments performed on HeLa cells, no cytotoxicity (94.3% cell viability) was found, even at the highest concentration tested (20 µM, 601.4 µg/mL), thus establishing the capability of the dendrimer reservoir (G4K) to cancel the cytotoxicity of the pristine UA (cell viability of 72.8% at 20 µM concentration).

Encouraged by the new physicochemical characteristics acquired by UA, the UA-G4K NPs obtained were biologically evaluated here, investigating their effects on both bacterial and normal eucaryotic cells. Particularly, a preliminary screening showed a remarkable and selective antibacterial activity against the *Enterococcus* genus. Therefore, we studied in detail the effects of UA-G4K NPs on several isolates of enterococci of different species, including in the study also strains of marine origin isolated from the seawater of the Ligurian coast, obtaining excellent results. This choice was inspired by the idea of comparing the sensitivity to UA-G4K NPs also of enterococcal strains that can be isolated in bathing waters.

Finally, to evaluate the feasibility of the clinical application of UA-G4K for the treatment of infections caused by VRE isolates of the *Enterococcus* genus [48], the cytotoxicity of UA-G4K NPs on human keratinocyte cells was evaluated. In parallel, G4K and UA were also tested under the same conditions for comparative purposes.

## 2. Materials and Methods

### 2.1. Chemicals Substances and Instruments Used in This Study

The biodegradable cationic dendrimer nanoparticles (NPs) loaded with UA (UA-G4K NPs) used in this study were recently synthesized according to the synthetic procedure reported by Alfei and collaborators [47]. Experimental details and characterization data are available in Appendix A. In addition, the experiments concerning the cytotoxicity of G4K, UA and UA-G4K toward eukaryotic ovarian cancer cells (HeLa) and the related results are available in Appendix A.

Chemical materials and methods, as well as instruments used for the physicochemical and biological characterization of UA-G4K NPs, are consultable in our previous work [47].

### 2.2. Microbiologic Investigations

#### 2.2.1. Description of the Microorganisms Considered in This Study

A total of 31 strains belonging to the Gram-positive and Gram-negative species were used in this study. All were clinical or marine isolates, belonging to a collection obtained from the School of Medicine and Pharmacy of University of Genoa (Italy), identified by VITEK^®^ 2 (Biomerieux, Firenze, Italy) or matrix-assisted laser desorption/ionization time-of-flight (MALDI-TOF) mass spectrometric technique (Biomerieux, Firenze, Italy). Two strains were of Gram-negative species (1 *Escherichia coli* and 1 MDR *Pseudomonas aeruginosa*). A total of 29 strains were Gram-positive, including two isolates of the genus *Staphylococcus* (1 MRSA and 1 MRSE) and 27 clinical isolates of the genus *Enterococcus*, including 6 vancomycin sensible (VSE) *E. faecalis*, 2 vancomycin resistant (VRE) *E. faecalis,* 1 VSE *E. faecium*, 4 VRE *E. faecium*, 3 *E. casseiflavus*, 3 *E. gallinarum*, 3 *E. avium,* and 2 *E. durans*. In addition, three strains of marine isolates of the genus *Enterococcus* were also included in this study (2 VRE *E. faecalis* and 1 VRE *E. faecium*).

#### 2.2.2. Definition of the Minimal Inhibitory Concentrations (MICs)

The antimicrobial activity of the substances considered in this study was investigated, determining their MICs. To this end, we followed the microdilution procedures specified by the European Committee on Antimicrobial Susceptibility Testing EUCAST [49] and described them in detail in our recent work [50].

#### 2.2.3. Time-Killing Experiments

Killing curve assays for UA and UA-G4K NPs were performed on representative isolates of *E. faecalis* and *E. faecium* species as previously reported [50,51]. Experiments were performed over 24 h at UA concentrations equal for all the strains (5 times the MICs, 10 µg/mL, 21.9 µM), and at UA-G4K NPs µM concentrations equal to those of UA (21.9 µM 658.4 µg/mL), to compare the potency of UA and UA-loaded NPs when used at the same dosage.

### 2.3. Evaluation of UA, G4K and UA-G4K NPs Cytotoxicity on Eukaryotic Normal Cells

#### 2.3.1. The Cell Culture Used in This Study

In these experiments, we used human skin keratinocytes cells (HaCaT) obtained and grown as described in a recent study published by us [50]. Briefly, HaCaT cells were obtained thanks to a generous gift from the Laboratory of Experimental Therapies in Oncology, IRCCS Istituto Giannina Gaslini (Genoa, Italy), and were tested and characterized at the time of experimentation as previously described [52].

#### 2.3.2. Assessment of Viability of HaCaT Cells Exposed to G4K, UA, and UA-G4K NPs

The viability assay was performed both on the empty dendrimer (G4K), UA and UA-G4K NPs following a procedure previously described [50]. Differently from our previous work [48], in this study, it was not necessary to process the data obtained from the viability essay by the principal component analysis (PCA) before using them to construct the curves of the cell’s viability percentages vs. the concentrations of samples experimented (0, 1, 5, 10, 15, 20, 25, 50, 75 and 100 μM).

### 2.4. Statistical Analyses

Concerning cytotoxicity studies, the statistical significance of differences between experimental and control groups was determined via two-way analysis of variance (ANOVA) with the Bonferroni correction. The analyses were performed with Prism 5 software (GraphPad, La Jolla, CA, USA). Asterisks indicate the following *p*-value ranges: * = *p* < 0.05, ** = *p* < 0.01, *** = *p* < 0.001. Concerning MIC values, experiments were made in triplicate, the concordance degree was 3/3 and ± SD was zero.

## 3. Results

### 3.1. Synthesis and Characterization of UA-G4K NPs

The procedure described in Section S1.1 and showed in Scheme S1 (SM) produced the UA-loaded fourth-generation dendrimer NPs (UA-G4K NPs) [47].

Their characterization by FTIR and NMR analyses, consultable in SM (Section S1.1), confirmed the UA-G4K structure [47]. The results of Scanning Electron Microscopy (SEM) experiments, performed to determine the morphology and average size of particles of UA-G4K, are available in Appendix A. The molecular weight (MW) of UA-G4K was calculated as described in Appendix A, while data concerning its water solubility are reported in Appendix A. The results of dynamic light scattering (DLS) analyses were performed to determine particle size (Z-ave, nm), polydispersity index (PDI), and Z-potential (ζ-*p*, mV) of UA-G4 NPs are reported in Appendix A. The experiments and the related results concerning the UA release profile, as well as the kinetics and mechanisms that govern the release of UA, are available in Appendix A. The cytotoxicity experiments on HeLa cells and relative results are in Appendix A. A detailed discussion of the characterization results is available in Alfei et al. (2021) [47]. To facilitate the readers, the main characteristics of UA-G4K NPs have been included in the following Table 1.

### 3.2. Antibacterial Properties of UA and UA-G4K NPs

Recently, we encapsulated UA, known for its in vitro antibacterial effects against strains of different Gram-positive genera [53,54,55,56], in the G4K dendrimer, proven free of antibacterial properties (results not reported). We obtained water-soluble UA-loaded NPs, not cytotoxic on HeLa cells. Before the herein study, it remained unknown if, after this nanomanipulation, the antibacterial properties of UA were maintained, increased, decreased, or changed in terms of target bacterial species. With the present research, we have addressed this shortcoming.

#### 3.2.1. Determination of MICs of UA-G4K and UA

Preliminary investigations were performed against representative strains of Gram-positive and Gram-negative species by determining the MIC values (MICs) of UA-G4K NPs. The MIC value of 128 µg/mL was assumed as the cutoff value above which the compound was deemed inactive, regardless of the corresponding µM concentrations, which in all determinations were for UA-G4K 8.1–65.2-fold lower than those of UA. The results showed that, as expected, due to the UA inactivity against Gram-negative species such as *E. coli* and *P. aeruginosa*, UA-G4K NPs were also inactive. Furthermore, the screening showed that, unlike free UA, UA-G4K was also poorly active against staphylococci (Table 2).

On the contrary, UA-G4K proved to possess remarkable antibacterial effects against clinical and marine isolates of the *Enterococcus* genus (Table 3).

As can be seen in Table 3, UA-G4K NPs were very potent against all clinical and marine isolates of the *Enterococcus* genus tested. In fact, except for *E. faecalis* 4 (MIC = 4.3 µM), the UA-G4K NPs displayed very low MICs (0.5–2.1 µM), which were 2.1–35.6-fold lower than those displayed by the untreated UA. This exciting result is due to the high DL% of UA-G4K (which resulted in the presence of 33 UA moles per mole of dendrimer) and to the favorable UA release profile from UA-formulation [47]. The high number of moles of UA that can be released from UA-G4K NPs allowed the UA-formulation we developed to exhibit antibacterial effects at very low molar concentrations. It should also be considered that, in hypothetical in vivo experiments, pristine UA would be difficult to administer without using harmful solvents and co-solvents, and, furthermore, its poor bioavailability could negatively affect its in vivo efficacy. On the contrary, nanoengineered water-soluble UA in the form of UA-G4K NPs could be easily administrable and suitable for biomedical applications. Meticulous literature research to find UA-based polymer formulations as possible antimicrobial agents revealed to us that, while there are numerous articles on the in vitro antimicrobial activity of unformulated UA or its derivatives, articles regarding the in vitro antimicrobial activity of UA-based polymer formulations are practically absent. Most of the previously prepared NPs loaded with UA have, in fact, been studied for their anti-inflammatory [57,58] or antitumor activity [59,60,61,62]. To the best of our knowledge, except for our previous article on UA-loaded dendrimer NPs whose antibacterial activity was not attributable to UA [55], the only article on the antimicrobial activity of a UA-liquid crystalline systems-based formulation established the improvement of the antifungal activity of UA [63]. Therefore, the present study is the first that concerns the successful evaluation of the antibacterial activity of water-soluble UA-enriched NPs.

#### 3.2.2. Relevance of Our Results

In this study, we have identified a new powerful antibacterial agent based on nanotechnologically modified UA (UA-G4K NPs), selectively active against different species of the *Enterococcus* genus, able to inhibit their growth regardless of their multi-resistance to existing antibiotics. Moreover, the physicochemical characteristics of UA-G4K [47], such as particles size, ζ-*p*-value, polydispersity index, UA release profile and kinetics, solubility in water, and stability in aqueous solution, suggest its suitability for any biomedical applications. As a result, UA-G4K may represent a possible new therapeutic option for addressing severe enterococcal infections caused by MDR species, which are of increasing concern around the world. Enterococci, in fact, are the predominant flora in the gut, which, from this biological district, can extend to the bloodstream or invade as colonizers other tissues, causing endocarditis, urinary tract infections, prostatitis, intra-abdominal infection, cellulitis, wound infection, or infections of the skin and soft tissues [64]. Enterococcal infections have become one of the most challenging concerns for physicians and researchers of our century due to the increasing prevalence of MDR strains and the drastic decrease in the activity of many of today’s antibiotics. One of the main issues in the treatment of severe enterococcal infections consists of the capability of these pathogens to be either tolerant or intrinsically resistant to a variety of antimicrobial agents. For example, *E. faecium* possesses an acetylase enzyme (AAC(6ʹ)) that causes higher MICs to aminoglycosides, such as tobramycin and amikacin, which negatively affects their synergistic effect with cell-wall agents [65]. Most *E. faecalis* strains have a gene (designated lsa) that encodes an ATP-binding protein that confers intrinsic resistance to quinupristin/dalforistin (Q/D) [66]. In the USA, *E. faecium* was found to be the most common enterococcal species isolated from immunocompromised patients [67]. While resistance to ampicillin is rare in *E. faecalis*, and vancomycin resistance is much less frequent, *E. faecium* resistance to antibiotics such as ampicillin (MIC = 128 µg/mL, 366.3 µM) and vancomycin (MIC = 512 µg/mL), which are used to be the cornerstone antibiotics for the treatment of enterococcal infections, is almost “normal” in hospital-associated isolates of *E. faecium*. Indeed, the infectious Disease Society of America (IDSA) has included *E. faecium* among the ESKAPE pathogens (where ESKAPE indicates *Enterococcus faecium*, *Staphylococcus aureus*, *Klebsiella pneumoniae*, *Acinetobacter* spp., *Pseudomonas aeruginosa*, and *Enterobacter* spp.) for which new therapies are urgently needed [48,68]. Against VRE enterococci (MIC of vancomycin against urinary VRE isolates = 512 µg/mL, 353.3 µM [69]), the use of very high concentrations of other existing antibiotics, such as ampicillin (128 µg/mL, 366.3 µM), whose MICs for *E. faecalis*, generally are 0.5 to 4.0 µg/mL (1.4 to 11.4 µM) [70], and for *E. faecium* are 4 to 8 µg/mL (11.4 to 22.8 µM) [70], have showed to counteract enterococcal urinary isolates in vitro [48,69]. Such an approach could be an alternative to the development of new antibacterial agents, but the most concern associated with this strategy is the consequent higher toxicity of antibiotics, when in vivo administered, due to their higher dosage. In this study, we demonstrated that the UA-G4K NPs that showed against VRE strains of *E. faecalis* and *E. faecium* (strains with * in Table 3) MICs in the range 0.5–1.1 µM, is 1.3–45.6-fold more potent than ampicillin against VSE strains of *E. faecalis* and *E. faecium.* Additionally, UA-G4K NPs were not only effective against VRE strains of both species against which vancomycin no longer works (MICs = 0.5–4.3 µM vs. MIC = 353.3 µM) but, according to a study also reporting the MIC of vancomycin against urinary VSE isolates of the *Enterococcus* genus (MIC = 1.4 µM), were in some cases more potent than vancomycin when effective [69].

#### 3.2.3. Curves from Time-Killing Experiments

To our knowledge, there is only one study in literature that established the bactericidal activity of both OA and UA on enterococci [71], while many articles reported only the MICs of the two triterpenoids, which despite being inactive against Gram-negative species, have shown remarkable antibacterial effects against Gram-positive species.

In particular, the authors stated that UA showed bactericidal activity on the *E. faecium* BM4147 strain when added to the culture medium at concentrations above 16 µg/mL (2 times MIC), while it only demonstrated bacteriostatic activity on *E. faecalis* at concentrations above 8 µg/mL (2 times MIC).

To confirm or disprove these results, we performed time-kill experiments with UA on different strains of *E. faecium* and *E. faecalis* species at concentrations 5× MICs. In this way, having the same MIC, all strains tested in these experiments were exposed to the same dose of UA (10 µg/mL, 21.9 µM), which is like the doses used in the time-kill experiments reported by Horiuchi [71]. To compare the potency of UA to that of UA-loaded nano-formulation (UA-G4K NPs), the same strains were exposed to equal micromolar concentrations of UA-G4K NPs. Figure 2 shows two representative killing curves obtained from experiments on *E. faecium* 21 (Figure 2a) and *E. faecalis* 4 (Figure 2b).

Overall, the results evidenced that, as reported in the study by Horiuchi [71], while UA was unequivocally bactericidal against *E. faecium* isolate, resulting in a log reduction > 3 after 24 h of exposure (Figure 2a), it was mainly bacteriostatic on *E. faecalis* 4. In fact, although against *E. faecalis* 4, it showed bactericidal effects after 3 h (4 logs reduction), steady regrowth began immediately and after 24 h, only a reduction of 2.2 logs was observed (Figure 2b). Concerning UA-G4K NPs, after 24 h of exposure, effects identical to those of UA were observed on both the clinical isolates *E. faecium* 21 and *E. faecalis* 4, and a similar profile in the activity was detected during the 24 h of exposure. Particularly, like UA, UA-G4K showed bactericidal effects against the *E. faecium* isolate, resulting in a > 3 logs reduction in the original cell number (Figure 2a), without observing any regrowth, while it was bacteriostatic against *E. faecalis* 4 (reduction in the number of cells of the inoculum < 3 logs) (Figure 2b).

Interestingly, these results differed from those observed for the lysine-modified fifth-generation cationic dendrimers G5K [72] and G5-PDK [50,73], which we have previously reported. In those studies, after a rapid decrease of > 4 logs in the original cell number (1–4 h), a consistent regrowth was observed. We justified bacterial regrowth by assuming a pH-dependent inactivation of cationic dendrimers through a process of self-degradation by an intramolecular amidation reaction [73]. As an example, which confirms our assumption, Hyldgaard et al., which used ε-poly-lysine (ε-PL) against *P. putida*, whose structure should undergo the same degradation process as assumed for our lysine-dendrimers, addressed this issue by administering a mixture of ε-PL/dextrin 50/50, where dextrin, which is a well-known stabilizer, certainly protected ε-PL from self-degradation [74]. In the present study, regrowth was not observed because, although cationic due to the presence of peripheral lysine moieties, as G5K and G5-PDK, G4K does not possess antibacterial properties, therefore its eventual inactivation did not affect the bactericidal properties of UA-G4K NPs, which are due to the delivering of UA not susceptible of inactivation.

### 3.3. Cytotoxicity Effects of G4K, UA and UA-G4K NPs on HaCaT Human Keratinocytes Cells

The solubility in water and an appropriately high value of the selectivity index (SI) are pivotal requirements to make a molecule worthy of consideration as a new therapeutic agent against bacterial infections. With our previous study [47], we have solved the drawbacks of UA related to water solubility, thus satisfying one of the first important needs. In this work, we have determined the SI of UA and of UA-G4 NPs, to establish the most suitable compound to be developed as a new antibacterial agent, also promising for future biomedical applications. To have an acceptable value of SI, new antimicrobial agents should ideally have low MIC values on bacteria coupled to a high lethal dose (expressed as LD_50_, which is the dose needed to kill 50% of the cells) on eukaryotic cells. In microbiology, the SI value is given by Equation (1) and provides a measure of the selectivity of the antimicrobial agent for bacteria.
(1)$$SI=LD_{50}/MIC$$

To determine the SI value for UA-G4K NPs and UA, we performed dose and time-dependent cytotoxicity experiments on human keratinocytes (HaCaT), and the results from the dose-dependent cytotoxicity experiments performed for 24 h were used to compute the LD_50_. The obtained LD_50_ and the MIC values were used to calculate the SI value of UA-G4K and UA against each isolate of the *Enterococcus* genus considered in this study and when compared, produced very satisfactory results.

#### Dose- and Time-Dependent Cytotoxicity Experiments

Dose- and time-dependent cytotoxicity experiments were performed to evaluate the effects of UA-G4K NPs on HaCaT keratinocytes cells. Cytotoxicity experiments under the same conditions were also performed for UA and G4K to evaluate the reciprocal effects on the original cytotoxicity of pristine UA and empty dendrimer. Such experiments were performed on HaCaT keratinocytes cells for several reasons. MDR isolates of the *Enterococcus* genus, which have been shown to be the preferred target of the antibacterial activity of UA-G4K NPs, are often the cause of a variety of skin, soft tissue, and wound infections [48]. Consequently, to assess the cytotoxicity of UA-loaded NPs, we selected human keratinocytes, which are the primary type of cell found in the epidermis, the outermost layer of the skin, and are more susceptible to colonization by pathogenic bacteria, fungi, parasites, and viruses. The cytotoxic activity of the samples, as a function of their concentrations (1–100 µM), was determined after 4, 12 and 24 h of exposure of the cells. The results were reported in Figure 3a–c.

As can be seen in Figure 3, for all compounds, the cytotoxic effects were both time- and dose-dependent. Particularly, after 4 h of exposure, at concentrations 5–100 µM, G4K was the less toxic compound, while UA-G4K NPs were slightly more toxic than UA up to 50 µM, showed the same cytotoxicity of UA at 75 µM, and were significantly less cytotoxic than UA at 100 µM (cells viability of 86.2% vs. 72.1%, respectively). Moreover, the cell viability was remarkably higher than 50% for all compounds, also at the higher concentration of 100 µM (96.4%, 72.1%, and 86.2% for G4K, UA and UA-G4K, respectively).

Differently, after 12 h of exposure, at concentrations 1–25 µM, G4K showed cytotoxicity comparable or slightly higher than those of pristine UA and UA-G4K NPs, except for the concentration 20 µM, at which (strangely) the more cytotoxic compound was UA-G4K (cells viability of 81.97% vs. 88.8% of G4K and 91.2% of UA). At concentration 50 µM, G4K was significantly more cytotoxic than UA and UA-G4K, while it proved a cytotoxicity significantly lower than that of UA at higher concentrations (cells viability of 56.1% vs. 50.7% at 75 µM and of 53.4% vs. 31.1% at 100 µM), thus establishing that UA was 1.7-fold more cytotoxic than G4K at 100 µM. Moreover, although showed comparable or lower cytotoxicity than that of two ingredients also at lower concentrations (except for 20 µM), evidencing their mutual contribution to reducing the cytotoxicity of the complex, at the concentrations of 75 and 100 µM, UA-G4K NPs were 1.5–1.6-fold less cytotoxic than G4K and 1.6–2.5-fold less cytotoxic than UA, leaving alive the 77% of cells also at the higher concentration tested (100 µM). As expected, the cytotoxic effects of all compounds were more marked after 24 h of cells exposure, but a similar trend to that observed for 12 h of treatment was maintained. Particularly, G4K was the more cytotoxic compound at concentrations 1–50 µM, at 75 µM, its cytotoxicity was higher than that of UA-G4K and comparable with that of UA, while at 100 µM, the more cytotoxic substance was UA (cells viability of 15.0% vs. 23.0% (G4K) and 52.0% (UA-G4K)). Moreover, while after 12 h of exposure to G4K, the viability of cells did not drop below 53.4%, also at 100 µM, after 24 h of exposure, cells viability was remarkably under 50% (36.9%) already at a concentration of 50 µM. At such concentrations, when exposed to untreated UA, cells viability remained higher than 50% (56.6%), but at greater concentrations, it dropped dramatically, reaching 26.8% at 75 µM and 15.0% at 100 µM.

Interestingly, when exposed to UA-G4NPs, cells viability remained higher than 50% (52 µM), also at 100 µM, which is 23.2–200-fold higher than the MICs determined on enterococci considered in this study. Finally, while the cytotoxic effects of UA-G4 NPs were comparable to those of UA up to concentration 25 µM, they were significantly lower at concentrations 50, 75 and 100 µM.

Collectively, by encapsulating UA in G4K, we realized the reciprocal reduction in the intrinsic cytotoxicity of the two ingredients when alone, achieving water-soluble UA-loaded NPs that showed potent antibacterial activity at not cytotoxic dosage. To have an idea of the selectivity of UA-G4K for bacteria, the SI (LD_50_/MIC) was determined, getting the LD_50_ from the data obtained from the cytotoxicity experiments at 24 h of exposure.

The LD_50_ and SI of UA were also determined for comparison purposes, while concerning G4K, only the LD_50_ was determined to compare its cytotoxicity with that of UA and UA-G4K since it was inactive as an antibacterial agent (not reported results).

For both G4K, UA, and UA-G4K, cell viability (%) was reported vs. concentrations (µM), and three dispersion graphs were obtained. The equations of the correspondent linear regression models obtained by the ordinary least squares (OLS) method were used to compute the respective LD_50_ (Figure 4).

Table 4 collects the three equations, the associated R^2^ values, the LD_50,_ and the range of SI for UA and UA-G4K; the SI values computed for each enterococcal isolate considered in this study are observable in Table 3.

The sufficiently high value of the coefficients of determination R^2^ (Table 4) ensured the linearity of the regressions.

Concerning the obtained LD_50_ values, although its LD_50_ was like that of pristine UA, the more lethal compound was the empty dendrimer G4K. Interestingly, the LD_50_ of the UA-loaded NPs was 2-fold higher than that of G4K and 1.8-fold higher than that of UA, confirming that by its nano-encapsulation, not only the water solubility of UA was remarkably improved, but also its cytotoxicity on HaCaT cells was considerably reduced.

In addition, if we consider the amounts of UA that the LD_50_ of UA-G4K can deliver after 24 h accordingly with its DL% and release profile (806.6 µg/mL, 1766.1 µM), the UA nanotechnologically manipulated resulted 32.2-fold less cytotoxic than untreated UA.

Moreover, considering the LD_50_ of untreated UA (54.9 µM) and the dose of UA-G4K necessary to deliver such amounts of UA (3.0 µM), it can be noted that at the obtained concentration of 3.0 µM, UA-G4K is practically not cytotoxic, leaving alive the 91.2% of cells. This means that the UA-loaded formulation obtained by merging UA and G4K, when will be administered at a dosage able to release a UA amount, which, if untreated, kills the 50% of HaCaT cells, will be lethal for only the 8.8% of exposed cells.

Concerning the SI values of UA (6.2–12.5) and UA-G4K (22.5–193.7), they were both much higher than those reported as acceptable to consider the new antibacterial agent suitable for therapeutic uses. Moreover, the SI values of the UA-G4K NPs were 3.6–31.2-fold higher than those of UA, thus establishing its higher suitability for biomedical applications and as a new therapeutic agent. In addition, considering that the UA released by the LD_50_ of UA-G4K should be 1766.1 µM, the SI of the nanoengineered UA was in the range 410.7–3532.2, that is 66.2–285.6-fold higher than that of untreated UA. Concerning the SI values that can be considered satisfactory, the reported opinions are conflicting. Some authors have hypothesized that SI values ≥ 10 make a molecule worthy of further investigation [75,76], while Weerapreeyakul et al. [77] proposed a lower SI value (≥3) to define a clinically applicable molecule as an anti-cancer agent. In microbiology, Adamu et al. [78,79] reported the antibacterial activities and SI of South African plant leaf extracts, and the most active extract showed an SI of 5.2. Famuyide et al. [80], who described the antibacterial activity of plant extracts on some Gram-positive and Gram-negative bacteria, stated that the extracts could be considered bioactive and non-toxic if SI > 1, while Nogueira and Estolio do Rosario reported that SI should not be less than 2 [81]. Due to these many diverging opinions on the SI acceptance criterion, we believe that further studies are needed to determine the minimum acceptable SI value. However, UA-G4K NPs, possessing SI values (24 h) 2.3–19.4-fold higher than the highest (10) reported as acceptable, can be considered suitable for clinical development.

## 4. Conclusions and Future Perspectives for UA-G4K NPs

In this study, water-soluble UA-loaded NPs (UA-G4K), previously obtained by UA encapsulation in an auto-biodegradable fourth-generation cationic dendrimer containing lysine (G4K), not possessing antibacterial properties, demonstrated remarkable antibacterial effects against several strains of different species of the *Enterococcus* genus, including VRE clinical and marine isolates of *E. faecalis* and *E. faecium.* The micromolar MIC values determined for UA-G4K NPs on 31 enterococcal strains were very low (0.5–4.3), regardless of the drug resistance patterns of the tested strains, thus satisfying the urgent need for new antibacterial agents active also against pathogens against which vancomycin is no longer effective. Furthermore, in all cases, the MICs of UA-G4K were lower than those determined for free UA, thus confirming that our innovative strategy, to encapsulate UA in G4K, not only succeeded in obtaining water-soluble UA formulations suitable for possible in vivo applications but also improved the antibacterial potency of the UA.

Confirming the observations reported for the first time by Horiuchi and co-workers and never confirmed until now, time-kill experiments performed with UA on representative isolates of *E. faecium* and *E. faecalis* species demonstrated that UA possesses bactericidal activity on *E. faecium* and bacteriostatic effects on *E. faecalis.*

Time-kill curves obtained with water-soluble UA-G4K NPs on the same strains showed that UA-loaded NPs, when administered at the same micromolar concentration of UA (21.8 µM), possessed a behavior identical to that of UA, exerting bactericidal effects on *E. faecium* isolates, while bacteriostatic effects on strains of *E. faecalis* species.

Enterococci, in addition to endocarditis, urinary tract infections, prostatitis, intra-abdominal infections, and concomitant bacteraemia, could also be responsible for serious infections of the skin, soft tissues, and wounds. Consequently, to evaluate the possible future clinical application of UA-G4K as a therapeutic agent against these infections, we evaluated its cytotoxicity on human keratinocyte cells (HaCaT).

The results showed that UA-G4K is practically non-cytotoxic to these cells and well tolerated, being its LD_50_ = 96.4 µM (4.4 times the dose that was bactericidal). Therefore, the selectivity indices calculated for UA-G4K NPs were very high and in the range 22.5–193.7. As already highlighted, if we compare equal micromolar concentrations of UA, G4K and UA-G4K, without considering the actual amount of UA that the prepared nano-formulation can deliver thanks to the entrapment of UA in G4K dendrimer, the original cytotoxicity of the two ingredients has been reduced by 1.8- and 2-fold, respectively, and the SI of UA was improved by 1.8–31.2 times.

Differently, if the actual amount of nanoengineered UA delivered by UA-nano-formulation is considered, the cytotoxicity of the untreated UA was decreased by 32.2 times, and the selectivity indices were improved by 32.9–569.7 times. With our study, we have detected a powerful antibacterial compound, highly selective for different species of the *Enterococcus* genus and non-cytotoxic toward mammalian cells. Overall, the novelty of our research consists in having successfully tested a new bactericidal agent based on UA potentially administrable in vivo because water-soluble and, consequently endowed with suitable bioavailability, free from cytotoxicity toward eukaryotic cells, and obtained exclusively with a nanotechnological approach which, differently from the nano-emulsion techniques, avoided the use of harmful solvent, co-solvents, and surfactants. To the best of our knowledge, currently, although there is so far no approved dendrimer drug in therapy, six dendrimer derivatives were reported in clinical trials, and seven are available on the market. Particularly, the following dendrimer-derived agents were reported to be in clinical trials: DEP^®^ docetaxel, DEP^®^ cabazitaxel and VivaGel^®^ (McGowan et al., 2011), a vaccine with the dendrimeric MAG-Tn3 for breast cancer, ImDendrim for inoperable liver cancer, and OP-101 for X-linked adrenoleukodystrophy, while Dendris, 3DNA^®^, Alert ticket^TM^, Polyfect^®^, Stratus CS^®^, VivaGel^®^ and Superfect^®^ are already present on the market.

In this regard, we believe that the UA-G4K NPs developed here can be considered for future clinical use. Indeed, we are confident that UA-G4K NPs may be suitable for oral administration since previous pharmacokinetic and pharmacodynamic studies on poly(amidoamine) (PAMAM) dendrimer-based drug formulations, administered orally for the treatment of hypercholesterolemia performed in Male albino Sprague-Dawley rats, showed suitable pharmacokinetic performances, even better than those of the suspension of the pure drug. In addition, several formulations of water-soluble drugs, obtained using dendrimers as solubilizing agents, showed better and suitable bioavailability.

## Data Availability

All data necessary to support reported results are present in the main text of the herein article and in the Appendix A.

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
