# Peer review of "Efficacy of Ursolic Acid-Enriched Water-Soluble and Not Cytotoxic Nanoparticles against Enterococci"

_pharmaceutics, 2021, doi:10.3390/pharmaceutics13111976_

Round 1
Reviewer 1 Report
COMMENTS TO THE AUTHOR(S)
This paper, titled "Efficacy of Ursolic Acid-Enriched Water-Soluble and Non-Cytotoxic
Nanoparticles Against Enterococci," describes the development of water-soluble Ursolic acid
(UA) loaded dendrimer nanoparticles (UA-G4K NPs) that are non-cytotoxic on HeLa 20 cells
and have promising physicochemical properties for clinical use. This research is interesting,
however, certain comments and suggestions need to be improved before it can be published.
1. In the abstract, page 1, line 16. The word "threat" doesn't seem to fit this context. replace
it with "threaten"
1. In the abstract, page 1, line 29. (22.4-192.8), please clarify?
2. Introduction: More extensive descriptions of the basic concepts of the development of
biocompatible and functional nanoparticles are required.
3. In the Materials and Methods section, please include descriptions of materials, such as
the manufacturer's city/state and country, as well as cell lines.
4. On what basis the concentration of UA was selected?
5. What is the loading efficiency of UA in the nanoparticles?
Author Response
COMMENTS TO THE AUTHOR(S)
This paper, titled "Efficacy of Ursolic Acid-Enriched Water-Soluble and Non-Cytotoxic
Nanoparticles Against Enterococci," describes the development of water-soluble Ursolic acid
(UA) loaded dendrimer nanoparticles (UA-G4K NPs) that are non-cytotoxic on HeLa 20 cells
and have promising physicochemical properties for clinical use. This research is interesting,
however, certain comments and suggestions need to be improved before it can be published.
1. In the abstract, page 1, line 16. The word "threat" doesn't seem to fit this context. replace
it with "threaten"
We agree with the Reviewer and thank him for his comment. As suggested, “threat” has been replaced with “threaten”. Please, see line 16, in the revised manuscript.
In the abstract, page 1, line 29. (22.4-192.8), please clarify?
We thank the Reviewer for his suggestion. The information has been clarified. Please, see line 29, in the revised manuscript.
Introduction: More extensive descriptions of the basic concepts of the development of
biocompatible and functional nanoparticles are required.
As requested by the Reviewer, an extensive description of the basic concepts of the development of biocompatible and functional nanoparticles and two additional references (Ref. 44-45) have been added in the Introduction. Please, see lines 99-117 and 131-132, and lines 735-738.
In the Materials and Methods section, please include descriptions of materials, such as
the manufacturer's city/state and country, as well as cell lines.
We apologise in advance with the Reviewer, but the information requested was already present in the original version of the manuscript.
Particularly, concerning materials and instruments used to prepare UA-G4K NPs, the related manufacturers, their city/state and country, as well as concerning the cell employed to assess the cytotoxicity of UA, G4K and UA-G4K NPs, they were reported in our previous article, cited in the present manuscript (Ref. 47, in the revised version of the manuscript), as notified in lines 164-166.
Concerning bacteria their description and origin is reported in section 2.2.1 (lines 168-180).
Concerning keratinocytes (HaCaT) their description and origin has been reported in our previous work (Ref. 50 in the revised version of the manuscript) as is reported in Section 2.3.1. Anyway, to satisfy the Reviewer, the origin of HaCaT cell has been repeated also in the present manuscript. Please, see lines 196-199.
On what basis the concentration of UA was selected?
The request of the Reviewer is unclear because he has not specified for what experiment.
Anyway, following we have tried to satisfy his request.
If the Reviewer refers to the cytotoxicity experiments, as specified in the manuscript (lines 413-416, 419-423), and in the caption of Figure 3, dose dependent experiments were conducted for all materials tested including UA, and concentrations in the range 0-100 µM were explored (lines 206 and 430-432). The range was chosen based on the MICs obtained in the microbiology experiments. Particularly, a concentration interval was chosen, that went from doses much lower and much higher than the MICs.
If the Reviewer refers to microbiology experiments, no dose was chosen, but the concentrations of UA-G4K NPs and UA, which were reported as active (MICs), were determined by the method of microdilution according to the EUCAST protocol as specified in the text (lines 181-185). The dose of UA for the time killing experiments were chosen by making 5 times the MICs as specified in the text (lines 187-192 and 359-366) and in the caption of Figure 2.
- What is the loading efficiency of UA in the nanoparticles?
We apologise in advance with the Reviewer, but the information requested was already present in the original version of the manuscript, both in Table 1 in the main text and in Table S1 in the Supporting Materials. Anyway, the DL% was 49.7±5.9%.
Reviewer 2 Report
The authors studied the efficacy of Ursolic acid-enriched water-soluble and not cytotoxic nanoparticles against gram-positive and negative bacterial strains.
In general, the article is written well, and the topic is interesting and timely.
The Introduction and Materials sections are written well providing all necessary details.
The Results section provides a detailed discussion about in vivo and in vitro experiments.
Comments:
The quality of English is very low. It concerns both grammar, orthography, and sentence structuring. I would strongly suggest improving the language (with the help of a native English speaker/user).
Regarding the design of the main experiments:
Why did the authors not use standard antibiotics (vancomycin, ampicillin, gentamicin, etc.) as negative/positive controls in cytotoxicity experiments?
To summarize, I would recommend accepting the manuscript after minor revision.
Author Response
The authors studied the efficacy of Ursolic acid-enriched water-soluble and not cytotoxic nanoparticles against gram-positive and negative bacterial strains.
In general, the article is written well, and the topic is interesting and timely.
The Introduction and Materials sections are written well providing all necessary details.
The Results section provides a detailed discussion about in vivo and in vitro experiments.
We thank the Reviewer for his appreciations.
Comments:
The quality of English is very low. It concerns both grammar, orthography, and sentence structuring. I would strongly suggest improving the language (with the help of a native English speaker/user).
We thank a lot the Reviewer, for his comment and suggestion, which has enabled us to improve the quality of our work. The manuscript has been revised by the native English teacher, Prof. Deidre Kants, who works for both the University of Genoa and of Pavia, which we have included in the acknowledgments in the revised version of our manuscript.
Regarding the design of the main experiments:
Why did the authors not use standard antibiotics (vancomycin, ampicillin, gentamicin, etc.) as negative/positive controls in cytotoxicity experiments?
We apologize in advance to the Reviewer, because, in our opinion, the use of standard antibiotics (as a positive or negative control) in cytotoxicity experiments, is unnecessary and makes no sense. To the best of our knowledge, the reference antibiotic should be used to have a positive/negative control in microbiology experiments, particularly where MICs are determined. In this regard, as the Reviewer can observe in Table 3, two reference antibiotic (ampicillin and vancomycin) were already present in the original version of the manuscript.
To summarize, I would recommend accepting the manuscript after minor revision.
We are very thankful to the Reviewer for his decision.
Reviewer 3 Report
Th manuscript “Efficacy of Ursolic Acid-Enriched Water-Soluble and Not Cyto-2 toxic Nanoparticles Against Enterococci” by Schito et al was found interesting with robust experimental design and established rational. I recommend publication subject to the following point:
- Demdrimers is being used in drug delivery research for more than 30 years but no sign of translation to clinic yet in foreseeable future – can authors comment on where do they see this research is going?
- Is this formulation to be suggested for oral delivery? Is it worth all these complications, would a nano-emulsion give a better bioavailability?
- Synthetic scheme with a representative draw of the product must be added to the paper
- The main aim is to improve solubility hence bioavailability- details solubility studies but be added and discussed not just one raw in a table.
Author Response
The manuscript “Efficacy of Ursolic Acid-Enriched Water-Soluble and Not Cyto-2 toxic Nanoparticles Against Enterococci” by Schito et al was found interesting with robust experimental design and established rational. I recommend publication subject to the following point:
Dendrimers is being used in drug delivery research for more than 30 years but no sign of translation to clinic yet in foreseeable future – can authors comment on where do they see this research is going?
We apologise in advance to the Reviewer, but we do not totally agree with his comment. In fact, already in 2008, one dendrimer-based Microbicide called VivaGel® was in clinical development as a topical gel used to prevent the spread of HIV and genital herpes [Majoros, I., Williams, C., & Baker Jr., J. (2008). Current Dendrimer Applications in Cancer Diagnosis and Therapy. Current Topics in Medicinal Chemistry, 8(14), 1165–1179. doi:10.2174/156802608785849049].
In addition, currently, six dendrimer derivatives were reported in clinical trials and seven are available on the market.
Particularly, the following dendrimer-derived agents were reported to be in clinical trials: DEP® docetaxel, DEP® cabazitaxel and VivaGel®, vaccine with the dendrimeric MAG-Tn3 for breast cancer, ImDendrim for inoperable liver cancer, and OP-101 for X-linked adrenoleukodystrophy, while Dendris, 3DNA®, Alert tickettm, Polyfect ®, Stratus CS®, VivaGel®, and Superfect® are already present on the market (International Journal of Pharmaceutics Volume 573, 5 January 2020, 118814).
Anyway, to satisfy the Reviewer, comments on this question have been added in the Conclusions Section (lines 572-582).
Is this formulation to be suggested for oral delivery? Is it worth all these complications, would a nano-emulsion give a better bioavailability?
Previously pharmacokinetic and pharmacodynamic studies of poly(amidoamine) (PAMAM) dendrimer-based drug formulations oral administered, for the treatment of hypercholesterolemia performed in Male albino Sprague-Dawley rats showed good pharmacokinetic performances, even better than those of pure drug suspension. Additionally, several drugs-formulations obtained by using dendrimers as solubilizing agents proved improved and suitable bioavailability (Chauhan, A., Anton, B., & Singh, M. K. (2020). Dendrimers for drug solubilization, dissolution and bioavailability. Pharmaceutical Applications of Dendrimers, 59–92. doi:10.1016/b978-0-12-814527-2.00003-2). Therefore, we are confident that also UA-G4K NPs, which possess cationic amine groups as PAMAMs, could be suitable for oral administration and could have good and suitable bioavailability.
Concerning the question if the use of a nano-emulsions (NE) would have a better bioavailability, we can answer to the Reviewer that, even if NE are characterized by improved solubility and bioavailability, surfactants, co-surfactants, and stabilizers are required (5–10%), which could be dangerous for humans and irritating for GIT (Polymers 2021, 13, 2262).
In addition, NEs stability is low, because the very small droplets initially obtained, may tend to re-aggregate along time with the formation and growth of undesired great crystals (Polymers 2021, 13, 2262).
We hope to have satisfied the Reviewer's requests. Anyway, additional comments concerning this question have been added in the Conclusions Section (lines 566-573 and 583-591).
Synthetic scheme with a representative draw of the product must be added to the paper.
The synthetic Scheme of the preparation of UA-G4K NPs is already present in the Supporting Materials (Scheme S1). Anyway, a synthetic Scheme also showing a representative draw of the product (UA-G4K NPs) have been included also in the main text (Scheme 1, line 135) to satisfy the Reviewer.
The main aim is to improve solubility hence bioavailability- details solubility studies but be added and discussed not just one raw in a table.
As reported prior to Table 1, referenced by the Reviewer, a detailed discussion of the physicochemical properties of UA-G4K NPs including solubility studies, is available in our previous work, cited in this manuscript as Ref 47. Please see line 229-232. Some additional comments have been included in the Conclusion Section.
This manuscript is a resubmission of an earlier submission. The following is a list of the peer review reports and author responses from that submission.